# Transcriptome-Guided Identification of Drugs for Repurposing to Treat Age-Related Hearing Loss

**DOI:** 10.3390/biom12040498

**Published:** 2022-03-25

**Authors:** Nick M. A. Schubert, Marcel van Tuinen, Sonja J. Pyott

**Affiliations:** 1Department of Otorhinolaryngology and Head and Neck Surgery, University Medical Center Groningen, University of Groningen, 9713 GZ Groningen, The Netherlands; n.m.a.schubert@umcg.nl (N.M.A.S.); m.van.tuinen@umcg.nl (M.v.T.); 2Graduate School of Medical Sciences Research School of Behavioural and Cognitive Neurosciences, University of Groningen, 9713 AV Groningen, The Netherlands

**Keywords:** hearing, age-related hearing loss, presbycusis, ageing, cochlea, RNA sequencing, drug repurposing, transcriptome

## Abstract

Age-related hearing loss (ARHL) or presbycusis is a prevalent condition associated with social isolation, cognitive impairment, and dementia. Age-related changes in the cochlea, the auditory portion of the inner ear, are the primary cause of ARHL. Unfortunately, there are currently no pharmaceutical approaches to treat ARHL. To examine the biological processes underlying age-related changes in the cochlea and identify candidate drugs for rapid repurposing to treat ARHL, we utilized bulk RNA sequencing to obtain transcriptomes from the functional substructures of the cochlea—the sensorineural structures, including the organ of Corti and spiral ganglion neurons (OC/SGN) and the stria vascularis and spiral ligament (SV/SL)—in young (6-week-old) and old (2-year-old) C57BL/6 mice. Transcriptomic analyses revealed both overlapping and unique patterns of gene expression and gene enrichment between substructures and with ageing. Based on these age-related transcriptional changes, we queried the protein products of genes differentially expressed with ageing in DrugBank and identified 27 FDA/EMA-approved drugs that are suitable to be repurposed to treat ARHL. These drugs target the protein products of genes that are differentially expressed with ageing uniquely in either the OC/SGN or SV/SL and that interrelate diverse biological processes. Further transcriptomic analyses revealed that most genes differentially expressed with ageing in both substructures encode protein products that are promising drug target candidates but are, nevertheless, not yet linked to approved drugs. Thus, with this study, we apply a novel approach to characterize the druggable genetic landscape for ARHL and propose a list of drugs to test in pre-clinical studies as potential treatment options for ARHL.

## 1. Introduction

Age-related hearing loss (ARHL), or presbycusis, is one of the most common conditions affecting older adults [1]. An estimated one in three adults aged 65 years and older have significant hearing loss [2], and, due to the growing size of the ageing population, is increasing in prevalence [3]. Presbycusis can greatly limit daily functioning and lead to loss of connectedness, social isolation, low quality of life, and depression [4,5]. Moreover, hearing loss in older adults is associated with an increased risk of cognitive impairment and dementia [6]. Despite the large numbers of adults affected by presbycusis and the multitude of comorbidities associated with presbycusis, approaches to treat ARHL have not advanced significantly over the past decades. Existing approaches rely on prevention [7], which has no universally established or effective strategies, and hearing aids, which do not completely restore all aspects of hearing [8].

The primary cause of presbycusis is age-related loss of function of the cochlea, the part of the inner ear involved in hearing. Cochlear ageing is complex and involves loss of cellular and molecular structures necessary for sensory transduction and neurotransmission, multiple cellular and molecular mechanisms, and both genetic and environmental factors [9]. Multiple studies have linked age-related loss of the sensory inner [10] and outer [11] hair cells present in the sensory epithelium, the organ of Corti (OC), and the primary auditory neurons, the spiral ganglion neurons (SGNs; [10]) to elevation of auditory thresholds and difficulty with speech recognition in noisy environments [11]. Moreover, previous work has documented age-related atrophy of the endocochlear potential (EP) generating substructures, including the stria vascularis (SV) and spiral ligament (SL), which contributes to hearing loss across frequencies [12]. A variety of cellular and molecular mechanisms have been implicated in age-related loss of cochlear functioning, including oxidative stress, inflammation, and apoptosis [13]. Finally, genetic factors, including syndromic and non-syndromic hearing loss mutations and common genetic variants in the population [9,14], as well as environmental factors, especially noise and ototoxins [9], contribute to ARHL.

Despite advances in our understanding of the pathophysiology of cochlear ageing as well as methods to deliver drugs locally to the cochlea [15], pharmaceutical treatments for presbycusis still do not exist. The lack of pharmaceutical treatments for ARHL is in stark contrast to the increasing number of drugs available to treat other age-related conditions. The few drugs that have reached clinical trials, based on an abundance of preclinical research, focus(ed) largely on mitigating oxidative stress but also on manipulating the cell death cascade and promoting hair cell regeneration [16,17]. The lack of approved drugs and nearly empty drug discovery pipeline indicate the urgency to utilize new strategies to identify drugs to treat presbycusis.

To this end, we developed a transcriptomics-based approach to investigate the biological processes underlying cochlear ageing and to identify potential drug targets to repurpose to treat cochlear ageing. Drug repurposing (or repositioning) is a strategy for identifying new indications for existing and approved drugs and can substantially shorten the drug discovery process from 10–17 years to 3–12 years [18,19]. We specifically hypothesized that changes in gene expression associated with age-related loss of cochlear function would vary between the sensorineural (e.g., OC and SGNs) and EP-generating (e.g., SV and SL) substructures of the cochlea given their distinct functions and, therefore, indicate distinct pharmaceutical strategies to treat presbycusis. Therefore, we obtained quantitative transcriptomes from these isolated cochlear substructures from young (6-week-old) and aged (2-year-old) mice. With these transcriptomes, we identified the genes differentially expressed with ageing within and between substructures and used this information to identify already approved drugs that are positioned to be repurposed to treat age-related loss of cochlear function, and hence presbycusis, safely and specifically. This work not only identifies drugs that should be prioritized for future pre-clinical investigation to treat presbycusis but also establishes a novel and urgently needed platform to feed the drug discovery pipeline and develop pharmaceutical treatments for presbycusis.

## 2. Materials and Methods

### 2.1. Animals

All experiments were approved (in protocol 1710324) by the animal ethics committee of the University of Groningen (UG) and University Medical Center Groningen (UMCG) and complied with guidelines for animal experiments from the UG/UMCG, the Netherlands, and European animal welfare law. A total of 6 C57BL/6 mice were used for these experiments: 3 mice aged 6 weeks and 3 mice aged 104 weeks (2 years). Mice were housed in a 12:12 h light:dark cycle and allowed *ad libitum* access to food and water. This strain of mice is known to have a mutation in the *Cdh23* gene, which results in accelerated ARHL at 16 kHz after 16 weeks (4 months) of age [20]. This pattern of hearing loss progresses with increasing age, with preservation of hearing function observed only in the lower frequencies by 54 weeks of age [21]. This pattern of high-frequency hearing loss matches the pattern of age-related hearing loss in humans. For these reasons, the C57BL/6 mouse has been used extensively to study presbycusis [22]. Due to the limited availability of 104-week-old mice, we examined hearing loss using auditory brainstem response (ABR) measurements in 74-week-old C57BL/6 mice obtained from the same breeding colony from which the 104-week-old mice used in this study were obtained. Wave I ABR responses, which reflect cochlear function, were only detectable at the lowest pure tone frequency tested (8 kHz) in mice aged 74 weeks and were significantly elevated compared to mice aged 6 weeks (74 weeks old: 85.0 ± 0.0 dB SPL, *n* = 4; 6 weeks old: 33.8 ± 3.0 dB SPL, *n* = 16; *p* < 0.05 Mann–Whitney test) [23].

### 2.2. Dissection of Cochlear Substructures and Isolation of RNA

To obtain cochlear tissues, mice were euthanized by decapitation while fully anaesthetized. Cochleae were dissected in ice-cold phosphate buffered solution (PBS). Cochlear tissue was micro-dissected into two substructures: the organ of Corti, including the spiral ganglion neurons (OC/SGN), and the stria vascularis, including the spiral ligament (SV/SL). Each replicate consisted of both OC/SGNs or both SV/SLs from one mouse. Thereafter, tissues were kept on ice and put in TRIzol reagent for RNA isolation, followed immediately by RNA extraction. A rotor-stator homogenizer (BioSpec Tissue-Tearor, Bartlesville, OK, USA) was used to disrupt the cellular structure of the tissues. The RNA extraction procedure followed the Arcturus picopure protocol, with the addition of a DNAase step. RNA quality and quantity was first checked with a ThermoFisher Nanodrop (Waltham, MA, USA). Samples were checked for RNA quality by performing capillary electrophoresis using a Perkin Elmer LabChip GX (Waltham, MA, USA). Samples with distinct 18S and 28S peaks, and RIN scores were chosen for sequence analysis. RNA samples that passed quality control were selected for sequencing analysis.

### 2.3. RNA Sequencing

RNA sequencing (RNA-seq) and quality control (QC) was performed by the Genome Analysis Facility (GAF) of the UMCG. Illumina TrueSeq RNA sample preparation kits were used to generate sequence libraries while using the Perkin Elmer Sciclone NGS Liquid Handler (Waltham, MA, USA). Obtained cDNA fragment libraries were sequenced on an Illumina HiSeq2500 (San Diego, CA, USA) (single reads 1 × 50 bp) in pools of multiple samples. Mus musculus.GRCm38 ensembleRelease 82 reference genome was used to align the trimmed fastQ files with hisat (https://github.com/infphilo/hisat, accessed on 20 March 2017). Sorting of aligned reads was performed using SAMtools (http://www.htslib.org/, accessed on 20 March 2017). The gene level quantification was performed by HTSeq and Ensembl version 82 (https://github.com/htseq/htseq, accessed on 20 March 2017) was used as the gene annotation database. FastQC (https://www.bioinformatics.babraham.ac.uk/projects/fastqc/, accessed on 20 March 2017) was used for quality control measurements of raw sequencing data. Picard-tools (https://broadinstitute.github.io/picard/index.html, accessed on 20 March 2017) calculated quality control metrics for aligned reads. Sequencing data was provided via a count table, which could be loaded into RNA-seq analysis software.

### 2.4. Identification of Differentially Expressed Genes (DEGs) and Functional Enrichment Analyses

Data were analyzed in R using existing RNA-seq data analysis workflows. Genes with no reads (0 counts in all samples) were excluded from the analysis. Differential expression analysis was performed using the DESeq2 package [24]. A false discovery rate (FDR) < 0.05 was used to determine whether genes were differentially expressed. This FDR was chosen based on best practices [24,25] and to maximize identification of potential therapeutic molecular targets. Genes were considered upregulated if the log_2_ fold change was above 0 and downregulated if below 0. Results were visualized using the following packages: tidyverse [26], ggplot2 [27], pheatmap [28]. Functional enrichment analysis was performed using the gProfiler2 package [29], using a g:SCS threshold < 0.05. Ontologies that were included in this analysis were: GO biological process, GO molecular function, GO cellular component, KEGG, Reactome, and WikiPathways.

### 2.5. Drug Identification

Information on drug targets was obtained using the Pharos database [30], DrugBank [31], and the dbparser package [32]. The Pharos database was used to quantify the “druggable” transcriptome. The list of DEGs per cochlear substructure was cross-referenced with the Pharos database, allowing identification of potentially suitable targets (gene products) for drug discovery. Drugs approved by regulatory bodies with single protein drug targets (e.g., specific) and no known ototoxic effects (e.g., safe) were selected from DrugBank. Up- and downregulated DEGs were cross-referenced with this list of drugs to identify potential drugs suitable for drug repurposing. To identify which biological processes and molecular functions were affected by the identified drugs, a network analysis of the targeted gene products (i.e., DEGs) was performed using GOnet [33]. The GOslim annotation function was used.

### 2.6. Curated Gene Lists

Genes involved in ageing and senescence were obtained from the Human Ageing Genomic Resources, a collection of databases and tools for studying the biology and genetics of ageing [34]. The GenAge database was used to identify genes in humans and mice (352 genes) that are related to longevity and/or ageing. The CellAge database was used to identify genes (279 genes) associated with cell senescence in human cell types. Finally, deafness genes identified in humans and mice (359 genes) were obtained from a previously curated dataset [35,36]. Curated gene lists are available in Appendix A.

## 3. Results

### 3.1. Cochlear Substructures Show Distinct Gene Enrichment

To first investigate the utility of transcriptomic analyses to identify biological processes associated with the sensorineural tissues, including the organ of Corti and spiral ganglion cells (OC/SGNs), and the endolymph-generating tissues, including the stria vascularis and spiral ligament (SV/SL), we compared transcriptomes between these two substructures isolated from young (6-week-old) mice (Figure 1A). When comparing the 1000 most abundantly expressed genes between each substructure (OC/SGN and SV/SL), approximately 70% were shared and approximately 30% were unique, consistent with both shared (likely housekeeping) processes as well as unique biological processes. When examining the 20 most abundantly expressed genes, genes among the most enriched in the OC/SGN include *Slc45a1*, *Otop1*, and *Fgf12*, which have been associated with neural tissues, and genes among the most enriched in the SV/SL include *Coro1a, Ska1*, and *Myo1g*, which are involved in cytoskeletal processes (Figure 1B).

To further identify and compare biological processes enriched in each cochlear substructures, we performed a functional enrichment analysis (Figure 1C). This analysis revealed a total of 246 and 445 significantly enriched ontologies in the OC/SGN and SV/SL, respectively, with approximately half of these enriched gene sets (*n* = 143) being shared between both substructures (Figure 1C, upper panel). The OC/SGN showed greater functional specialization in neuronal-associated processes, whereas the SV/SL showed greater enrichment of processes associated with metabolism and the immune function (Figure 1C, lower panel). The shared biological processes include vesicle-mediated transport, intracellular signaling transduction, and regulation of cell communication. The complete list of enriched ontologies is available in Appendix A. These results show that transcriptomic profiles identify both shared and unique biological processes associated with the distinct cochlear substructures and that, not surprisingly, neuronal genes and processes are enriched in the sensorineural structures and metabolic genes and processes are enriched in the endolymph-generating structures.

### 3.2. Cochlear Substructures Show Distinct Age-Related Changes in Gene Expression

We were next interested in investigating the age-related changes in gene expression within and between the two cochlear substructures. Comparison of replicates by heatmap and hierarchical clustering analysis, in which samples are plotted based on pairwise Pearson’s correlation analysis (Figure 2A), and by principal component analysis (PCA), in which the two principal components that explain most of the variance are plotted (Figure 2B), both show clear clustering of replicates by substructure and age. PCA revealed higher variance among replicates from substructures isolated from older animals. Previous work has indicated that higher variance in gene expression in older samples is biologically relevant [37,38]. When comparing variance among substructures, 73.5% of the variance arises from substructure differences whereas only 5.3% of the variance arises from age-related differences between substructure replicates, indicating that transcriptomic differences between substructures are greater than transcriptomic differences arising from ageing. Because of the clear clustering of replicates, subsequent analyses included transcriptomes from all replicates.

Differential gene expression analysis revealed both unique and shared changes in gene expression in response to ageing when comparing the OC/SGN and SV/SL (Figure 2C). A total of 583 genes are significantly differentially expressed with ageing in the OC/SGN, and a total of 821 genes are significantly differentially expressed with ageing in the SV/SL (Figure 2C, upper panel). Ageing resulted in slightly more downregulation of gene expression in the OC/SGN (up: *n* = 243 genes; down: *n* = 340 genes) and slightly more upregulation of gene expression in the SV/SL (up: *n* = 496 genes; down: *n* = 325 genes). Only a small proportion of these differentially expressed genes (DEGs) were shared between cochlear substructures (20.4% of all OC/SGN DEGs and 14.5% of all SV/SL DEGs). When examining the top 20 up- and downregulated genes in response to ageing in the OC/SGN and SV/SL (Figure 2C, lower panel), there is no overlap in genes. The most upregulated gene in the OC/SGN is *Otol1*, which encodes OTOL1, a protein important for biomineralization and associated with vestibular disorders [39]. The most downregulated gene in the OC/SGN is *Mab21l2*, which encodes a nuclear protein linked to neural development and is enriched in SGNs in the base of the cochlea [40]. Another significantly downregulated gene in the ageing OC/SGN is *Calb2*, which encodes CALB2, and is enriched in the type Ia SGNs [40,41]. The most upregulated gene in the SV/SL is *Cxcl13*, which encodes a chemokine that was previously identified as upregulated in the ageing cochlea [42]. Our results suggest that upregulation with ageing is specifically attributed to increased expression in the SV/SL substructure. The most downregulated gene in the SV/SL is *Ppef1*, which encodes a serine/threonine protein phosphatase for which the substrate and vertebrate function are unknown (see [43]). A complete list of DEGs is available in Appendix A.

### 3.3. Cochlear Substructures Show Distinct Changes in the Expression of Genes Associated with Ageing, Senescence, and Deafness as Well as Gene Enrichment with Ageing

The largely non-overlapping DEGs identified in the ageing OC/SGN and SV/SL (Figure 2C) suggest that distinct biological processes underly age-related changes between these two cochlear substructures. To investigate these processes further, we first compared genes differentially expressed with ageing in each substructure to curated gene lists. Specifically, we compared DEGs to (1) genes related to longevity and/or ageing in mice and humans (the GeneAge database in the Human Ageing Genomic Resources [34]); (2) genes associated with human cellular senescence (the CellAge database in the Human Ageing Genomic Resources; [34]), a collection of molecular pathways associated with degeneration and ageing in various organisms and linked to age-related loss of cochlear function [44]; and (3) genes associated with deafness in mice and humans [35,36], many of which have been linked to presbycusis [45,46]. We identified DEGs belonging to each of these curated datasets in each of the cochlear substructures (Figure 3A). However, there were largely nonoverlapping lists of DEGs (<30%) shared between the cochlear substructures and each of the three curated datasets (Figure 3A). Specifically, 3 (all upregulated in both substructures), 1 (upregulated in both substructures), and 7 (all downregulated in both substructures) shared DEGs (listed in Figure 3A) were differentially expressed in both substructures and identified in either the curated ageing, senescence, or deafness gene lists, respectively. These findings suggest that although the cochlear substructures show age-related changes in transcriptomic profiles that overlap broadly with genes linked to ageing, senescence, or deafness, the individual genes, and likely the underlying mechanisms, are largely unique between substructures. The complete list of curated genes is available in Appendix A.

To identify the biological processes associated with ageing in each of the two cochlear substructures, we performed functional enrichment analyses. This analysis revealed a total of 187 and 214 sets of genes significantly enriched with age in the OC/SGN and SV/SL, with many of these enriched gene sets (>70% or 151 gene sets) being shared between both substructures (Figure 3B, upper panel). Notably, functional enrichment of these differentially expressed gene sets reveals that in the OC/SGN neuronal associated processes are downregulated with age (Figure 3B, lower panel). In the SV/SL, neuronal-associated processes as well as morphology-associated processes are downregulated with ageing (Figure 3B, lower panel). The downregulation of neuronal associated processes could be attributed to the gene ontologies used in this analysis, which includes a very broad set of genes. Both substructures show upregulation of immune response processes with age. Together, these analyses show that genes broadly associated with ageing, senescence, and deafness are also differentially expressed with ageing in the cochlear substructures. However, several other genes are also differentially expressed with ageing, reflecting cochlear-specific age-related changes.

### 3.4. Transcriptomic Analysis Identifies Substructure-Enriched and Depleted Targets Suitable for Drug Repurposing

To identify drugs that could be repurposed to treat age-related loss of cochlear function specifically and safely, we took advantage of the DrugBank database [31]. We first identified drugs that target the protein products of genes differentially expressed with ageing in each cochlear substructure. This initial list of drugs was reduced to those that were approved, specific, and safe. Specific drugs were identified as drugs that have a single target. Drugs with reported ototoxic effects were excluded. Furthermore, we searched for antagonists (also including inhibitors, binders, antibodies, or degradation) that target upregulated genes or agonists (also including activators) that target downregulated genes. This search yielded a total of 27 specific and safe drugs (Figure 4). These drugs targeted products of genes uniquely differentially expressed with ageing exclusively in either the OC/SGN (12 drugs) or SV/SL (15 drugs). Most drugs target products of DEGs that are significantly upregulated with ageing. Potentially specific and safe drugs that can be repurposed to treat age-related loss of cochlear function include activators, inhibitors, antagonists, agonists, binders, antibodies, and degraders. These drugs target a variety of gene products linked to biological processes that are likely impacted by age-related loss of cochlear function, including neurotransmission (e.g., GABRA1), ionic homeostasis (e.g., ATP1A1), cell adhesion (e.g., ITGAL and ITGB3), and inflammation (e.g., IL4R).

### 3.5. Potential Drug Targets Link Several Biological Pathways and Functional Mechanisms

The gene products targeted by the drugs identified in our analysis (Figure 4) are involved in a large network of molecular mechanisms. To identify the shared mechanisms targeted by these drugs, we performed a gene network analysis to investigate the interactions between the targeted gene products. For this analysis, we used a specific set of GO slim gene sets. These GO slim ontologies are broader gene sets and, therefore, provide a more comprehensive overview of the processes associated with the genes included. With this approach, we found that many of the gene product targets are involved in general biological processes (Figure 5A), including extracellular matrix organization, transport, and lipid metabolic process. Moreover, even on the network edges, which represent processes connected with a lower number of genes, we identified biological process involved with ageing, immune system, and nervous system. When investigating molecular function ontologies (Figure 5B), we found that various processes, including ion binding, ATPase activity, oxidoreductase activity, DNA binding, and cytoskeletal protein binding are involved (Figure 5B). These results show that the gene products targeted by these drugs impact important pathophysiological processes.

### 3.6. Characterization of the Druggable Genetic Landscape for Treatment of Age-Related Loss of Cochlear Function

Our analysis revealed 27 specific and safe drugs potentially repurposed to treat presbycusis. We also wanted to investigate the druggable landscape revealed by age-related transcriptomic changes more broadly. To this end, we cross-referenced our list of products encoded by the identified DEGs with the Pharos database to assess the distribution of drug target development levels [30]. The Pharos target development levels fall into four categories: (1) clinical (T_Clin_): drug targets linked to at least one approved drug; (2) chemical (T_Chem_): proteins with established chemistry but no known link to approved drugs; (3) biological (T_Bio_): proteins that have a confirmed Mendelian disease phenotype and GO annotations but otherwise do not fall into either the T_Clin_ or T_Chem_ categories; (4) dark genome (T_Dark_): proteins that do not meet the criteria for the other three categories [30,47]. Cross-referencing indicated a variety of potential drug targets (products of DEGs) that fall into the T_Clin_ and T_Chem_ categories and, therefore, can be targeted by either already approved drugs or known active ligands (Figure 6). Slightly smaller fractions of T_Clin_ and T_Chem_ annotated targets (DEGs) were found in the OC/SGN compared to the SV/SL (T_Clin_: *n* = 16 (4%) in the OC/SGN versus *n* = 33 (6%) in the SV/SL; T_Chem_: *n* = 30 (8%) in the OC/SGN versus *n* = 68 (13%) in the SV/SL). Most annotated targets were classified in the T_Bio_ category (*n* = 305 (78%) in the OC/SGN and *n* = 373 (71%) in the SV/SL), indicating that these biological targets are well-studied (i.e., referenced in literature, have GeneRIF annotations, antibodies and molecular or biological function data, and associated phenotypes) and, therefore, are amenable to drug target development. A smaller fraction of potential targets in both the OC/SGN (*n* = 38 (10%)) and SV/SL (*n* = 50 (10%)) fell into the T_Dark_ category, for which there is very limited knowledge to support drug target development. In short, age-related transcriptomic change in the cochlear substructures indicate a highly druggable landscape but also urgent need to identify more drugs.

## 4. Discussion

### 4.1. Overview

In this study, we analyzed transcriptomes from the sensorineural (OC/SGN) and metabolic (SV/SL) cochlear substructures isolated from young (6-week-old) and aged (2-year-old) mice to identify the genes and biological processes differentially expressed between substructures and with age-related loss of cochlear function. Consistent with their distinct functions, we found that the sensorineural and metabolic substructures are transcriptionally distinct, with the OC/SGN showing enrichment of transcripts associated with neuronal processes and the SV/SL showing enrichment of transcripts associated with processes related to metabolism and the immune system (Figure 1).

These substructures show both unique and overlapping transcriptional changes with ageing (Figure 2 and Figure 3). With ageing, both substructures show upregulation of transcripts that are involved in ageing and senescence and downregulation of identified deafness genes (Figure 3). When examining differential expression more broadly, genes involved in processes related to the immune system were upregulated in both substructures (Figure 3). In the OC/SGN, there is more age-related downregulation of transcripts related to neuronal processes. In the SV/SL, there is more downregulation of morphology-associated processes.

Based on these age-related transcriptional changes, we compiled a list of 27 approved drugs that are suitable to be repurposed to treat age-related loss of cochlear function (Figure 4). The drugs identified target protein products that are differentially expressed with ageing uniquely in either the OC/SGN or SV/SL. Most of the identified drugs target the protein products of genes that are upregulated with ageing and uniquely differentially expressed in the SV/SL. These drugs target the protein products of genes, interrelate diverse biological processes (Figure 5), including neuronal processes, inflammation, oxidative stress, and cell signaling, many of which have been previously associated with age-related loss of cochlear function [13]. Cross referencing with a recent, comprehensive review of therapeutic approaches to treat inner ear and central hearing disorders [48], indicates that the drugs identified in this study are not currently under investigation for therapeutic use.

### 4.2. Drugs Poised for Repurposing to Treat Age-Related Loss of Cochlear Function

**Drugs targeting neuronal processes.** Two major groups of drugs target the products of genes involved in neuronal processes and differentially expressed (upregulated) with ageing in the OC/SGN. The first gene, *Atp1a1*, encodes the Na,K-ATPase alpha1 subunit, which is expressed in supporting cells that express the glutamate transporter GLAST and likely contributes to uptake of glutamate from the afferent synaptic cleft [49]. Expression of Na,K-ATPases is extensively controlled, from transcription to post-translation [50], and both mineralocorticoids and glucocorticoids are known to activate transcription *Atp1a1* [51]. Age-related changes in steroid hormone signaling, may be responsible for the upregulation of *Atp1a1* in the OC/SGN with ageing; steroid hormone signal was not identified in the functional enrichment analysis. We identified multiple drugs that inhibit the Na,K-ATPase alpha1 subunit, including digoxin, acetyldigitoxinn, deslanoside, bretylium, digitoxin, and almitrine. Suppression of the Na,K-ATPase alpha1 subunit may treat presbycusis by increasing glutamate in the synaptic cleft and enhancing afferent synaptic transmission. The Na,K-ATPase alpha1 subunit is also expressed in the marginal cells of the stria vascularis and fibrocytes of the spiral ligament [52]. We saw no age-related changes in *Atp1a1* expression in the SV/SL, although reduced expression of the Na,K-ATPase (correlated to reduced endocochlear potential) has been documented in a gerbil model of age-related hearing loss [53]. Importantly, the drugs we identified in this study are not associated with hearing loss typical of some Na,K-ATPase-targeting loop diuretics [54]. However, and as discussed further below, functional studies are needed to determine if suppressing the Na,K-ATPase alpha1 subunit is effective in treating presbycusis, especially given its co-expression in both substructures of the cochlea.

The second gene involved in neuronal processes and differentially expressed (also upregulated) with ageing in the OC/SGN is *Gabra1,* which encodes the alpha1 subunit of the ionotropic GABA_A_ receptor. The GABA_A_ alpha1 subunit has been identified in the spiral ganglion neurons of the cochlea and is involved in the modulation of afferent neurotransmission via release of GABA from the lateral efferent system [55]. Ethchlorvynol, methohexital, zaleplon, and brexanolone inhibit the GABA_A_ alpha1 subunit. Various lines of evidence (reviewed in [55]) suggest that GABA_A_ receptors inhibit afferent neurotransmission. Thus, drugs that inhibit the GABA_A_ alpha1 subunit may treat presbycusis by reducing lateral efferent inhibition and thereby enhancing afferent neurotransmission.

**Drugs targeting inflammation.** Inflammation has been extensively associated with age-related hearing loss [13,56]. We identified multiple genes that are differentially expressed (upregulated) in the SV/SL and involved in the inflammatory response, including *Alox5*, *Itgal*, *Itgb3*, *Elane, Pik3cd*, *Il4r*, *Ptgs1*, and *Prg2*. Inhibiting the products of these genes may treat presbycusis by reducing inflammatory responses. *Alox5* encodes a lipoxygenase involved in the synthesis of leukotrienes. This gene product is targeted by zileuton, an antileukotriene synthesis inhibitor, currently approved to treat mild-to-moderate chronic asthma. Inhibition of leukotrienes, which mediate vasoconstriction, may specifically treat presbycusis by increasing cochlear microcirculation [57]. Moreover, leukotriene receptor inhibition has been previously shown to reduce noise-induced permanent threshold shifts and hair cell loss [58].

Cell adhesion is an important part of the inflammatory response. Both *Itgal* and *Itgb3* are genes involved in cell adhesion. *Itgal*, which encodes the integrin alpha L chain, is upregulated in the cochlea after noise exposure [59]. *Itgb3*, which encodes the integrin beta chain beta 3, has been previously associated with idiopathic sudden sensorineural hearing loss [60]. The gene products of *Itgal* and *Itgb3* are targeted by, respectively, lifitegrast, approved to treat dry eye, and eptifibatide, which inhibits platelet aggregation and is approved to manage myocardial infarction.

The genes *Elane*, *Pik3cd*, and *Il4r* have also been previously associated with noise-induced and age-related hearing loss. *Elane* encodes neutrophil elastase, which is involved in chronic inflammation and inhibited by the alpha-1-proteinase inhibitor [61]. Upregulation of neutrophil elastase is observed following noise-induced hearing loss [62]. *Pik3cd* encodes a phosphoinositide 3-kinase that is involved in the immune response, has been previously associated with idiopathic sudden sensorineural hearing loss [63], and is targeted by the drug idelalisib, approved to treat certain blood cancers. *Il4r* encodes the interleukin-4 receptor, which has been associated with age-related hearing loss in a genetic population study [64]. The interleukin-4 receptor plays a pivotal role in the inflammatory response and is targeted by the drug dupilimumab, a monoclonal antibody approved to treat allergic diseases such as eczema, asthma, and nasal polyps.

Finally, two genes, *Ptgs1* and *Prg2*, are associated with inflammatory responses but not previously associated with hearing loss. *Ptgs1* encodes a prostaglandin-endoperoxide synthase that also functions as a cyclooxygenase and peroxidase and, therefore, may be involved not only in inflammatory responses but also oxidative stress [65,66]. The *Ptgs1* protein product is approved for use an antipyretic and analgesic drug. Finally, *Prg2*, encodes a proteoglycan that forms a major component of the eosinophil granule. The protein product is targeted by the drug chrymopapain, which is approved, but now discontinued, to treat disc herniation.

**Drugs targeting oxidative stress.** Oxidative stress has also been associated with age-related hearing loss [13,67]. We identified two genes, *Xdh* and *Pgd*, that are differentially expressed (upregulated) in the SV/SL and involved in oxidative stress. *Xdh* encodes xanthine dehydrogenase, which belongs to the family of oxidoreductases, and is inhibited by the drug allopurinol. Allopurinol is approved to decrease urate levels and is frequently used in the treatment of chronic gout, which is associated with a higher risk of hearing impairment in the elderly [68]. Allopurinol has been shown to protect against cisplatin-induced and noise-induced hearing loss in animal models [69,70]. Another drug, febuxostat, also targets xanthine dehydrogenase and is approved for the management of chronic hyperuricemia in cases where there is an inadequate response or intolerance to allopurinol. Although febuxostat has a similar mechanism of action as allopurinol, the effectivity of febuxostat in protecting against acquired hearing loss has not been verified. *Pgd* encodes a phosphogluconate dehydrogenase, which belongs to the family of oxidoreductases, and is important in protecting cells from oxidative damage by concomitantly generating NADPH [71]. Mice overexpressing a related protein, glucose-6-phosphate dehydrogenase, showed better age-related preservation of auditory thresholds [72]. Gadopentetic acid, approved as a gadolinium-based MRI contrast agent, also inhibits erythrocyte phosphogluconate dehydrogenase [73].

**Drugs targeting cell signaling.** We identified two genes that are differentially expressed with ageing and liked to cell signaling. *Map2k1* is upregulated with ageing in the OC/SGN, and *Pth1r* is downregulated with ageing in the SV/SL. *Map2k1* encodes mitogen-activated protein (MAP) kinase kinase. Although *Map2k1* has not been directly implicated in age-related hearing loss, MAP kinases, in general, integrate multiple cellular signaling pathways biochemical signals and are, therefore, poised to interrelate various biological processes involved in age-related hearing loss [74]. The protein product of *Map2k1* is inhibited by cobimetinib, approved to treat unresectable or metastatic melanoma. *Pth1r*, encodes the parathyroid hormone 1 receptor, which mediates activity via G protein-mediated activation of adenylyl cyclase, the major cell signaling molecules. The protein product of *Pth1r* can be agonized by abaloparatide and teriparatide, both approved for the treatment of osteoporosis. Osteoporosis has been associated with elevated auditory thresholds in a large population cohort study [75]. Countering osteoporotic effects might have a protective effect on age-related decline of hearing thresholds.

**Drugs targeting other biological processes.** We also identified genes that are differentially expressed with ageing in the SV/SL and linked to other, diverse biological process. *Srebf1* encodes the sterol regulatory element binding transcription factor 1 (SREBP1) and is upregulated with ageing in the OC/SGN. SREBP1 is inhibited by omega-3-acid ethyl esters, approved to reduce triglyceride levels in adults with severe hypertriglyceridemia. Two independent lines of evidence suggest that inhibition of SREBP1 may be useful to treat presbycusis. First, in another transcriptomic investigation of the mouse cochlea, *Srebf1* showed greater suppression of expression following noise exposure in the noise-resilient type 1a SGNs compared to the more noise-sensitive type Ib and Ic SGNs [76]. Second, inhibition of SREBP1 activation shows neuroprotective effects in an in vitro model of stroke [77]. Two other genes, *Pdgfra* and *Abcc9*, are downregulated with ageing in the SV/SL. *Pdgfra* encodes platelet-derived growth factor receptor alpha, which is targeted by olaratumab, an antibody approved to treat certain types of soft tissue sarcoma. *Pdgfra* is expressed in the developing cochlea [78] and has been shown to be slightly downregulated in type IB and type II SGNs [76]. *Abcc9* encodes a member of the superfamily of ATP-binding cassette (ABC) transporters that forms ATP-sensitive potassium channels [79]. The *Abcc9* protein product is activated by nicorandil, approved to treat angina pectoris. Human loss-of-function mutations in the *ABCC9* gene are associated with mild sensorineural hearing loss [80,81]. Nicorandil may treat presbycusis by activating ATP-sensitive potassium channels in the SV/SL.

### 4.3. Limitations of This Study

A transcriptomic approach to identify the biological processes associated with cochlear ageing and prioritize candidate drugs for repurposing to treat ARHL is both powerful and promising. Nevertheless, limitations of this study should also be recognized. First, the animal model used in this study poses certain limitations. We examined C57BL/6 mice; a widely used strain long studied as a model for ARHL [22]. The cause of ARHL in this strain has been linked to mutations in *Cdh23* gene [82]. Mutations in *Chd23* are linked not only to ARHL in mice but have also more recently been linked to presbycusis in humans [45,46]. Nevertheless, examination of different mouse strains and animal models, which show variations in the patterns and processes of age-related loss of cochlear function, are necessary to investigate more comprehensively the biological processes contributing to ARHL as well as better distinguish changes in gene expression linked specifically to ARHL and not ageing per se. Furthermore, we investigated transcriptional changes in very old (2-year-old) mice to identify changes associated with complete (terminal) ARHL. Future studies should include additional younger timepoints to identify changes associated with earlier (and perhaps more treatable) stages of ARHL.

Second, the transcriptomic approach used in this study poses certain limitations. In our study, transcriptomic changes were used as a proxy for proteomic changes, although the two are not perfectly correlated [83]. Future studies should aim to integrate both transcriptomic and proteomic approaches. Additionally, we examined transcriptional changes in the functional substructures of the cochlea. Our findings would be complemented by future studies that additionally investigate age-related changes in specific cell types as has recently been done to examine transcriptional changes in the inner ear in response to noise exposure [76]. Finally, when identifying drugs, we restricted our search to inhibitors of protein products encoded by genes that were upregulated with ageing and activators of protein products encoded by genes that were downregulated with ageing. However, the underlying pathophysiology is not revealed by the direction of gene expression changes. Moreover, only the protein products of differentially expressed genes were considered. The identified drugs may also target protein products of genes expressed but not differentially expressed with ageing in the other substructure. Thus, functional studies are necessary to determine whether an inhibitor or activator of a potential drug target is necessary to treat presbycusis and to examine potential substructure-specific effects [84].

### 4.4. Outlook and Future Directions

In this study, we presented novel insights into the biological processes involved in cochlear ageing. Moreover, we provided a list of targets and drugs suitable for drug repurposing to treat presbycusis. Thus, this work enables researchers to design experiments to test the efficacy of these candidate drugs to treat ARHL as part of preclinical studies using in silico, in vitro, and in vivo approaches. Importantly, the biological processes that underly ARHL overlap with other forms of hearing loss, including noise-induced hearing loss and otoxocity [85,86]. Therefore, the candidate drugs identified in this study may be useful to treat other forms of acquired hearing loss. Finally, our study further revealed that a large fraction (approximately 80%) of the genes differentially expressed with ageing in both substructures encode protein products that are promising drug target candidates but are, nevertheless, not yet linked to approved drugs (Figure 6). This finding indicates that the druggable landscape to treat presbycusis is promising but almost entirely untapped. Thus, pharmaceutical treatment of presbycusis is achievable but needs to leverage new approaches, similar to the one outlined in this study, as well as interdisciplinary cooperation to not only repurpose existing drugs but fill the drug discovery pipeline.

## Figures and Tables

**Figure 1 biomolecules-12-00498-f001:**
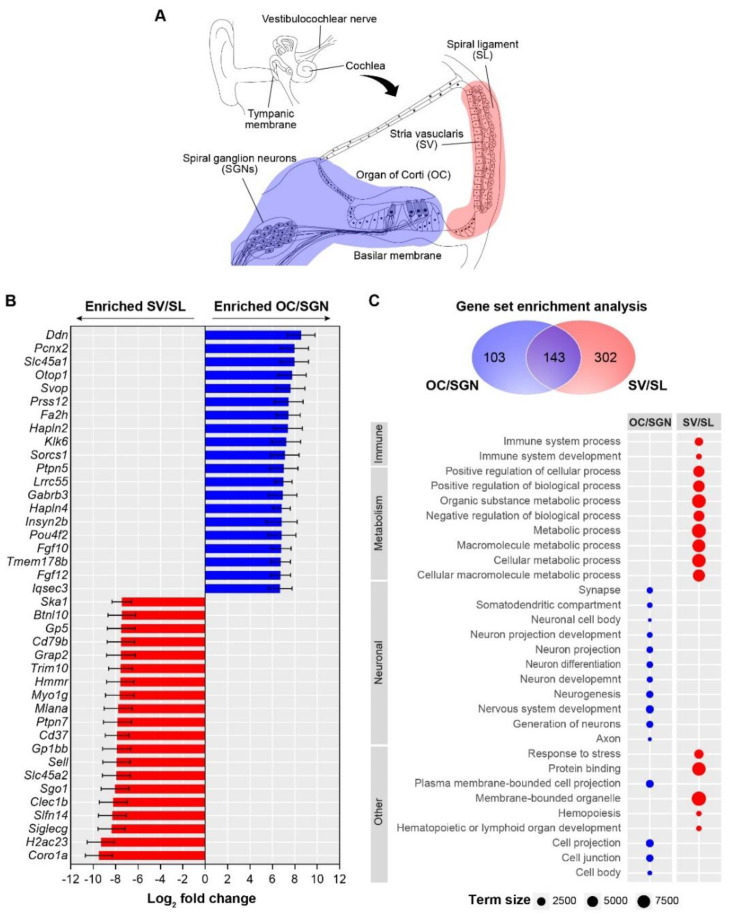
**Cochlear substructures show distinct gene enrichment consistent with their distinct biological processes.** (**A**) Transcriptomes were obtained from the two cochlear substructures: the sensorineural structures (blue), which included the organ of Corti together with the spiral ganglion neurons (OC/SGN), and the metabolic structures (red), which included the stria vascularis with the spiral ligament (SV/SL). (**B**) Differential expression analysis comparing cochlear substructures isolated from young mice reveals specific genes that are enriched in either the OC/SGN (blue bars) or the SV/SL (red bars). The 20 genes most enriched in each substructure are shown. (**C**) Gene set enrichment analysis revealed both unique and overlapping gene sets associated with specific biological processes (gene ontologies) in the OC/SGN (blue) and SV/SL (red). The top enriched biological processes are different between the OC/SGN (blue circles) and SV/SL (red circles). Term size is the number of genes included in the gene ontology. In some cases, genes defining the ontology are not expressed in the cochlear substructures. A complete list of enriched processes is available in Appendix A.

**Figure 2 biomolecules-12-00498-f002:**
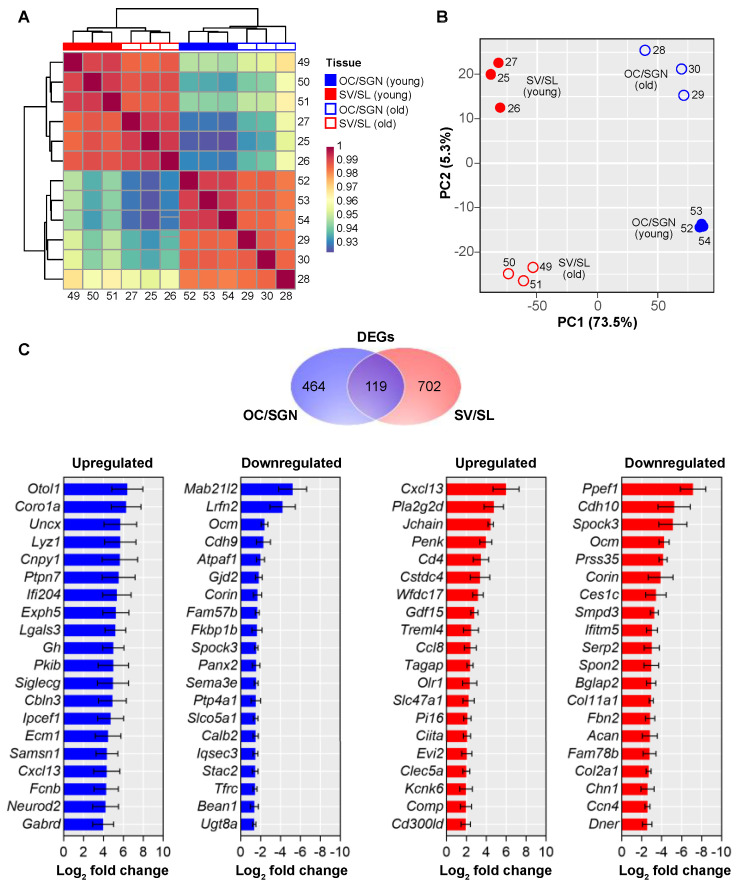
**Cochlear substructures show distinct age-related changes in gene expression** (**A**) Hierarchical clustering heatmap of Pearson correlation coefficients reveals clustering between cochlear substructure replicates. The legend indicates replicate identities. (**B**) Principal component analysis reveals clustering of replicates by substructure. The largest variance (PC1 or 73.5%) separates replicates by substructure (OC/SGN versus SV/SL), whereas separation by age (young versus old) accounts for a much smaller fraction of the variance (PC2 or 5.3%). Replicate identities are indicated. (**C**) A Venn diagram of genes differentially expressed with ageing (DEGs) shows that there are both unique and overlapping DEGs between the OC/SGN (blue) and the SV/SL (red). The 20 genes most up- and downregulated with ageing in each substructure are shown. The complete list of DEGs is available in Appendix A.

**Figure 3 biomolecules-12-00498-f003:**
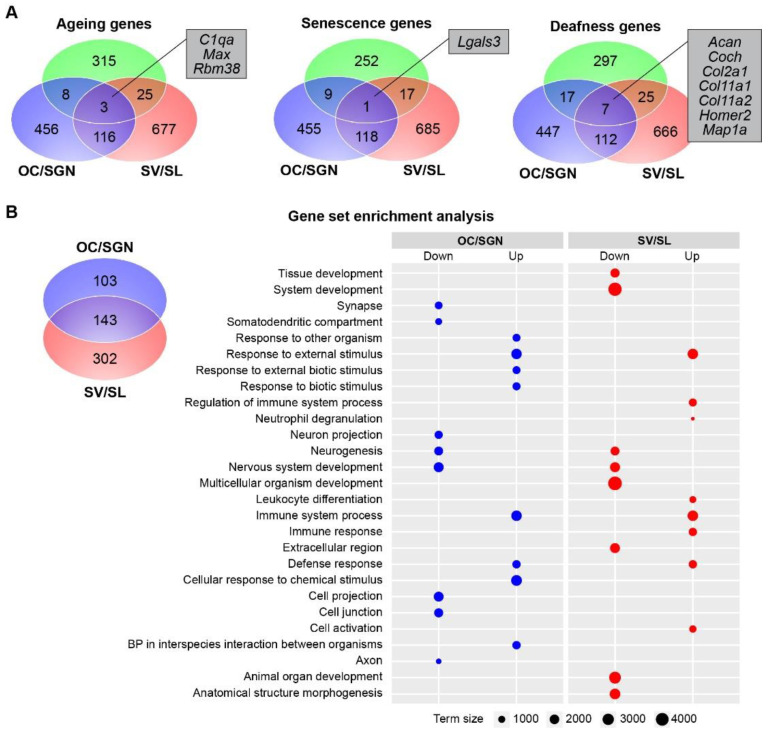
**Cochlear substructures show distinct changes in the expression of genes associated with ageing, senescence, and deafness as well as gene enrichment with ageing**. (**A**) Venn diagrams of the genes differentially expressed with ageing in either the OC/SGN (blue) and SV/SL (red) and belonging to curated datasets of either ageing, senescence, or deafness genes (green) were identified in each cochlear substructure although the overlap of genes between substructures (listed) was small. Of these overlapping genes, ageing and senescence genes are upregulated in both substructures and deafness genes are downregulated in both substructures. (**B**) A Venn diagram of the gene set enrichment analysis reveals both unique and overlapping gene sets associated with specific biological processes (gene ontologies) in the ageing OC/SGN (blue) and SV/SL (red). Separate enrichment analysis of pathways for up- and downregulated genes revealed the biological processes most enriched with ageing in the OC/SGN (blue circles) and SV/SL (red circles). These processes are mostly non-overlapping. A complete list of enriched processes is available in Appendix A. Term size is the number of genes included in the gene ontology. In some cases, genes defining the ontology are not expressed in the cochlear substructures.

**Figure 4 biomolecules-12-00498-f004:**
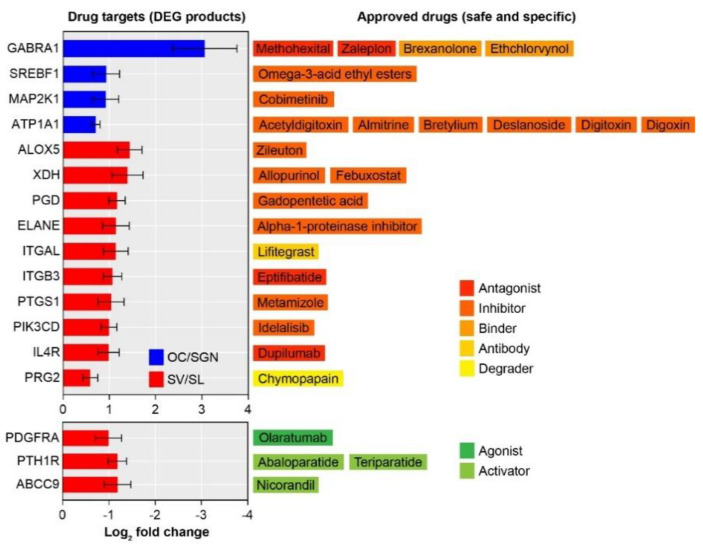
**Transcriptomic analysis identifies substructure-enriched and depleted targets suitable for drug repurposing.** The products of DEGs and their log_2_ fold changes associated with ageing in the OC/SGN (blue) and SV/SL (red) are plotted together with the specific and safe drugs targeting those gene products. The mechanism of action for each drug is indicated.

**Figure 5 biomolecules-12-00498-f005:**
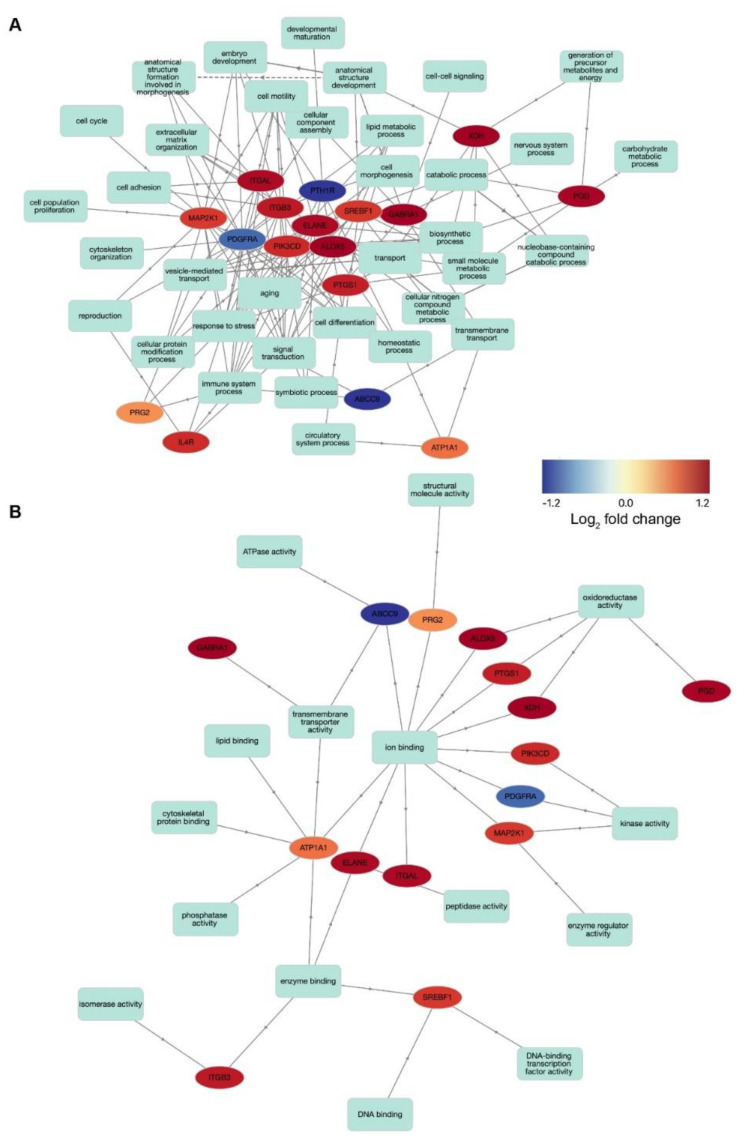
**Potential drug targets link several biological pathways and functional mechanisms** Network analyses identifying the interactions between the differentially expressed gene (DEG) products (drug targets) involved in (**A**) biological processes and (**B**) molecular functions using gene ontologies containing a subset of the terms (i.e., GOslim). The color map represents the log_2_ fold change values with ageing. Unconnected DEGs are not shown.

**Figure 6 biomolecules-12-00498-f006:**
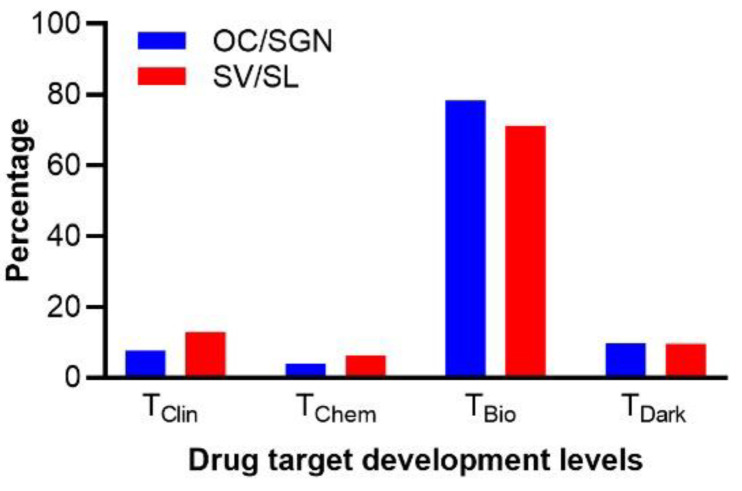
**Transcriptomic analysis reveals the druggable landscape for the treatment of age-related loss of cochlear function.** The drug target development levels of the products encoded by the differentially expressed genes (DEGs) associated with ageing in the OC/SGN (blue) and SV/SL (red) were determined using the Pharos database. The relative percentages per substructure per level are plotted as a function of level.

## Data Availability

RNA sequencing data is available on the University of Maryland gene Expression Analysis Resource (gEAR) [87]. https://umgear.org/p?s=3ea6634f accessed on 21 March 2022.

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
