# Peer review of "Transcriptome-Guided Identification of Drugs for Repurposing to Treat Age-Related Hearing Loss"

_biomolecules, 2022, doi:10.3390/biom12040498_

Round 1

Reviewer 1 Report

This manuscript reports the analysis of transcriptomes for the identification of drugs for repurposing to treat age-related hearing loss. The study was performed in a mice model, analyzing the transcriptomes from the sensorineural (OC/SGN) and metabolic (SV/SL) cochlear substructures isolated from young (6-week-old) and aged (2 year-old) mice. This work has performed by bulk RNA sequencing to obtain transcriptomes from the functional substructures of the cochle the sensorineural structures. The patterns of gene expression and gene enrichment between substructures and with ageing was revealed by transcriptomes, and the age-related transcriptional changes, i.e. the protein products of genes differentially expressed with ageing in DrugBank, led to the identification of the proposed drugs, which may be used to treat age-related hearing loss. Experiments in this work were well performed, and the manuscript is well written. This work is useful for future investigators in this field, and it is recommended publication.

Author Response

We would like to thank the Reviewer for his or her efforts in reviewing this manuscript. We appreciate the positive comments about our work and agree this work is useful for future investigators.

Reviewer 2 Report

The authors used C57BL/6 mice as a model for ARHL and compared the transcriptome in cochlea substructures between young mice with old mice to see molecular causes for the aging related morbidity and check potential drugs that can target these differentially expressed genes. The article is well-written. But there are some major flaws in the study. First, how do you know there is hearing loss in the old mice group? Is there any behavior change? You need to show some measurements that support the difference of hearing wellness between the two groups. Second, you cutoff value for the logFC of DEGs is too low. Normally, logFC needs to be at least 1 for a DEG. How can you judge difference based on 0? There must be some false positive ones. I strongly advice you to check proteins expression of some important druggable genes you listed in Figure 4 and mentioned in the discussion. Some Western blotting experiments showing corresponding changes in those molecules are needed to support your conclusion.

Author Response

Comments reviewer 2 

The authors used C57BL/6 mice as a model for ARHL and compared the transcriptome in cochlea substructures between young mice with old mice to see molecular causes for the aging related morbidity and check potential drugs that can target these differentially expressed genes. The article is well-written.But there are some major flaws in the study. First, how do you know there is hearing loss in the old mice group? Is there any behavior change? You need to show some measurements that support the difference of hearing wellness between the two groups.

We appreciate the Reviewer’s effort critically reviewing our manuscript. Regarding the comments on auditory function of these mice, especially in the two-year-old group, there is extensive literature on the irreversible, age-related changes in hearing function in the C57BL/6 mouse strain. This mouse strain is often used as a model of human age-related hearing loss studies. A comprehensive study by Ison et al. provides a longitudinal analysis of hearing thresholds in C57BL/6 mice aged between 10 and 54 weeks (see Figure 1)1. The results of this study show that at 54 weeks of age, these mice have severe high-frequency hearing loss with only some preservation of auditory function in the lower frequencies1. This pattern of age-related hearing loss closely matches patterns observed in humans. Based on these findings and the extensive previous characterization of hearing loss in this mouse strain, we have no doubt that C57BL/6 mice aged 104 weeks also have hearing loss. Although for reasons explained above (Response to Comments Academic Editor) we are not able to characterize hearing loss in mice aged 104 weeks, we do now include results from C57BL/6 mice aged 74 weeks to validate hearing loss in mice from the same colony from which mice on our experiments were obtained. This new text reads (lines 93-106): “This strain of mice is known to have a mutation in the Cdh23 gene, which results in accelerated ARHL at 16 kHz after 16 weeks (4 months) of age [20]. This pattern of hearing loss progresses with increasing age, with preservation of hearing function observed only in the lower frequencies by 54 weeks of age [21]. This pattern of high-frequency hearing loss matches the pattern of age-related hearing loss in humans. For these reasons, the C57BL/6 mouse has been used extensively to study presbycusis [22]. Due to the limited availability of 104-week-old mice, we examined hearing loss using auditory brainstem response (ABR) measurements in 74-week-old C57BL/6 mice obtained from the same breeding colony from which the 104-week-old mice used in this study were obtained. Wave I ABR responses, which reflect cochlear function, were only detectable at the lowest pure tone frequency tested (8 kHz) in mice aged 74 weeks and were significantly elevated compared to mice aged 6 weeks (74 weeks old: 85.0 ± 0.0 dB SPL, n = 4; 6 weeks old: 33.8 ± 3.0 dB SPL, n = 16; p < 0.05 Mann Whitney test) [23].” We hope this addition sufficiently addresses the reviewer’s concerns.

Second, you cutoff value for the logFC of DEGs is too low. Normally, logFC needs to be at least 1 for a DEG. How can you judge difference based on 0? There must be some false positive ones.

Different log2foldchange cut-off values are used in the literature, including the cut-off value we used in this investigation2. As the Reviewer correctly indicates, using tighter log2foldchange or false discovery rate cut-off values would decrease the likelihood of identifying false positives but would also increase the likelihood of excluding positive results. Due to the urgency of identifying candidate therapeutic molecular targets, we believe that, in this exploratory analysis, it is preferential to identify as many potential targets as possible, and, therefore, choose to use these cut-off values. To better justify our approach, we added new text (lines 141-143): “This FDR was chosen best on best practices [24,25] and to maximize identification of potential therapeutic molecular targets.” We hope the reviewer appreciates these considerations.

I strongly advice you to check proteins expression of some important druggable genes you listed in Figure 4 and mentioned in the discussion. Some Western blotting experiments showing corresponding changes in those molecules are needed to support your conclusion.

We entirely agree with the Reviewer that analyses of changes in protein expression would validate the transcriptional changes identified in this study. As already mentioned in the manuscript, we acknowledge that transcriptomic and proteomic expression is not perfectly correlated and that additional, protein validation is necessary. (See lines 572-579: “the transcriptomic approach used in this study poses certain limitations. In our study, transcriptomic changes were used as a proxy for proteomic changes, although the two are not perfectly correlated [82]. Future studies should aim to integrate both transcriptomic and proteomic approaches.”) Unfortunately, for reasons explained above (Response to Comments Academic Editor), western blotting experiments to confirm transcriptional changes are simply not possible. We also believe that these experiments are not necessary as part of this exploratory study and are instead better investigated using advanced techniques (including high-throughput proteomic technologies, like mass spectrometry) in future studies. We hope the reviewer appreciates these considerations.

Additional references:

  1. Ison, J.R.; Allen, P.D.; O’Neill, W.E. Age-Related Hearing Loss in C57BL/6J Mice Has Both Frequency-Specific and Non-Frequency-Specific Components That Produce a Hyperacusis-Like Exaggeration of the Acoustic Startle Reflex. Journal of the Association for Research in Otolaryngology 2007, 8, 539–550, doi:10/c628bw.
  2. Munnamalai, V.; Sienknecht, U.J.; Duncan, R.K.; Scott, M.K.; Thawani, A.; Fantetti, K.N.; Atallah, N.M.; Biesemeier, D.J.; Song, K.H.; Luethy, K.; et al. Wnt9a Can Influence Cell Fates and Neural Connectivity across the Radial Axis of the Developing Cochlea. J Neurosci 2017, 37, 8975–8988, doi:10/gbw74q.

Round 2

Reviewer 2 Report

I have no more comments on this paper.